# Links between Couples’ Cynical Hostility and Mental Health: A Dyadic Investigation of Older Couples

**DOI:** 10.3390/bs14040283

**Published:** 2024-03-28

**Authors:** Dikla Segel-Karpas, Roi Estlein, Ashley E. Ermer

**Affiliations:** 1Department of Gerontology, University of Haifa, Haifa 3498838, Israel; 2School of Social Work, University of Haifa, Haifa 3498838, Israel; restlein@univ.haifa.ac.il; 3Department of Family Science and Human Development, Montclair State University, Montclair, NJ 07043, USA; ermera@montclair.edu

**Keywords:** cynical hostility, couples, marital relations, anxiety, depression

## Abstract

Whereas sharing a life with someone with high cynical hostility can be straining, little is known about how partner’s cynical hostility is associated with one’s mental health. In this paper, we report the findings from a longitudinal dyadic study using two waves of a large and representative American sample of older adults and their spouses to examine how one’s own and their spouse’s cynical hostility longitudinally affect anxiety and depressive symptoms. Results from APIM analyses suggest that both husbands’ and wives’ anxiety and depressive symptoms were negatively associated with their own cynical hostility, both within each time point and longitudinally. Partners’ cynical hostility, however, predicted only husbands’ mental health cross-sectionally. Furthermore, a moderating effect was identified, although it was not consistently observed across all analyses. Specifically, when a partner’s cynical hostility was high, the association between one’s own cynical hostility and their mental health was stronger, especially for women. Theoretical and practical implications are discussed.

## 1. Introduction

Cynical hostility is a social-cognitive schema suggesting that people are a source of wrongdoing and are motivated solely by selfish goals [1]. This basic lack of trust in others’ intentions can leave individuals vulnerable to poor social relationships [2]. Previous studies identified cynical hostility as a risk factor for loneliness [2], perceived disrespect from others [3], and problematic family relationships [4]. However, relatively little is known about how mental health is affected when sharing life with a partner high in cynical hostility. The current manuscript addresses this gap in the literature by describing the links between own and partner’s cynical hostility and the experience of anxiety and depressive symptoms. Utilizing two waves of a large and representative sample of older couples we examine how one’s own and a spouse’s self-reported cynical hostility are associated with both one’s own and their partner’s mental health.

### 1.1. Cynical Hostility

Cynical hostility is described as a social-cognitive schema, suggesting that other’s motives are selfish, and hence their intentions and good deeds cannot be trusted [1]. Cynicism is a component of general hostility, which is defined as an “attitude toward others, consisting of enmity, denigration, and ill will” [5] (p. 26). Interestingly, cynical hostility is considered a personality trait [6], developing in early parent–child interactions [7] and remaining relatively stable over one’s life course [8]. Nevertheless, social environmental factors, such as socioeconomic status, were found to be associated with cynical hostility [9]. A large body of research has linked cynical hostility to various health problems, which are especially prominent in older adulthood. This line of research has primarily focused on cardiovascular diseases [1,10], but has also linked cynical hostility with pulmonary function [11] and all-cause mortality [12,13]. Several mechanisms are theorized to link cynical hostility with poor health outcomes. First, the basic lack of trust suggests hypervigilance in social interactions that prompts the activation of the physiological stress reaction [1]. Second, individuals with higher levels of cynical hostility report less supportive and more conflictual social networks, which, in turn, undermines their ability to cope with potential stressors [14,15]. Finally, research suggests that cynical hostility is linked to more hazardous health behaviors, leading to greater disease susceptibility [16].

Cynical hostility has also been linked to mental health, with longitudinal studies suggesting that cynical hostility is predictive of loneliness and depressive symptoms over time [2,16,17,18]. Similar to the mechanisms linking cynical hostility to physical health, it is suggested that poor social support, as well as stress and frequent conflicts in social relationships, mediate the link between cynical hostility and mental health [2,4,17,18]. Furthermore, hostility has also been linked, albeit less consistently, to anxiety [17]. Some studies [18], but not others [19], suggest that anxiety mediates the hostility–depression link. In this study, we examine the associations between cynical hostility and depression and anxiety symptoms, focusing on one’s own and marital partner’s cynical hostility as possible predictors.

### 1.2. Cynical Hostility in Close Relationships

The interpersonal theory of personality [20] is an appropriate framework for understanding how the effects of cynical hostility on social relations unfold. According to this theory, personality is defined as a pattern of interpersonal situations that characterize an individual. Individuals’ expectations from others and their behavior in social situations elicit responses that correspond with their initial expectations. This reciprocal process leads to the formation of a relatively stable set of social behaviors [21,22]. The negative expectations of those with high levels of cynical hostility are expressed in behavior, and, in turn, shape social partners’ reactions in what was termed a ‘vicious cycle’ of hostility [3,22].

Studies indicate that cynical hostility takes a toll on social relationships. For example, in a series of studies, cynical hostility and disrespect [3] and cynical hostility and loneliness [2] were predictive of one another. In addition, individuals with high levels of cynical hostility report less social support and more social conflicts [14,23,24,25]. Interestingly, individuals with high levels of cynical hostility also provide less support to others [26] and manage to benefit less from the support offered to them [27]. Moreover, individuals with high levels of cynical hostility find it more difficult to self-disclose, which can undermine intimacy [28].

Cynical hostility also plays a role in shaping family relationship dynamics. In the family unit, emotions and behaviors are easily transmitted between members [29]. In a study of older parents, cynical hostility was found to shape older parents’ relationship with their adult children [25]. Studies on cynical hostility within couples suggest that high levels of cynical hostility are associated with poorer relationship outcomes. For example, in a longitudinal sample following younger adults for a period of 11 years, a high level of cynical hostility was associated with a greater likelihood to remain single, be divorced, or separated [30]. Within the marital dyad, men with higher levels of cynical hostility reported lower levels of relationship satisfaction compared to men with lower levels of cynical hostility [31]. Both men and women with higher levels of cynical hostility also reported higher marital conflict [23]. Cross-spousal effects were found in a different study [32], suggesting that both one’s own and their partner’s cynical hostility are associated with high marital conflict and low marital quality. Finally, focusing on mental health, men’s cynical hostility was associated with their own and their spouse’s depressive symptoms [33], whereas wives’ cynical hostility was associated with their own and their husband’s loneliness [4].

There are several reasons to assume that one’s own and their partner’s cynical hostility will be associated with one’s depressive and anxiety symptoms. At the core of cynical hostility lies great vulnerability [34]. In the marital dyad, this vulnerability, coupled with a sense of isolation, can increase distress in several different ways: First, the negative cognitive schema of distrust can be transmitted from one partner to another [35], leaving both, independently, feeling more vulnerable. Second, because individuals with high levels of cynical hostility may feel uncomfortable in situations that require self-disclosure [36], cynical hostility may inhibit achieving emotional intimacy within the intimate relationship, leaving partners feeling lonely [37]. Third, because individuals with high levels of cynical hostility feel uncomfortable in social interactions, they may discourage dyadic social interactions. Others may also avoid the couple because cynical hostility is not well accepted in social situations. Both these processes can leave one’s partner isolated from others [4]. Finally, these processes can be more prominent in older adulthood. The experience of marital conflict may have a wear and tear effect on one’s mental health [38], increasing a sense of burnout, depressive symptoms, and anxiety as couples age together. These effects can be especially meaningful when both partners have high levels of cynical hostility, as they can act to enhance one another’s negative outlook on the social environment, increase marital tension, and leave both partners socially isolated [2,4,39]. Support for this proposition can be found in studies addressing dyadic neuroticism, suggesting that when both members are high in neuroticism, there is an additive toll on their marital relations [40,41]. Furthermore, according to the socioemotional selectivity theory [42], as people age, they tend to focus on emotionally, rather than instrumentally, meaningful social goals, suggesting that a greater emphasis is given to close bonds, such as family relations, close friends, and one’s spouse. This focus inward may have negative implications if one or both partners are high in cynical hostility, and they can find it difficult to establish secure relations with another person and with people close to them [4].

Based on the reviewed literature suggesting that not only one’s own and their partner’s cynical hostility can lead to greater vulnerability to poor mental health, but that also that own and partner’s cynical hostility have an additive effect to this association, our hypotheses are as follows: 

**H1.** 
*Both one’s own and their partner’s cynical hostility will be positively associated with individuals’ anxiety and depressive symptoms.*


**H2.** *Partner’s cynical hostility will moderate the association between one’s own cynical hostility and anxiety and depressive symptoms, such that higher levels of their partner’s cynical hostility will strengthen the association*.

Because cynical hostility, as well as mental health, has been correlated with social environmental factors [9], we also considered features such as race, age, and self-rated health as covariates in our models.

## 2. Materials and Methods

### 2.1. Participants

Data were derived from the 2006 (T1) and 2010 (T2) waves of the Health and Retirement Study (HRS). The HRS is a nationally representative sample of adults aged 50 or older in the United States. Spouses of respondents, regardless of age, are also included in the HRS. In order to test our hypotheses, we utilized the lifestyle and psychosocial questionnaire (the ‘Leave Behind’). This questionnaire is administered every four years to a subset of participants. This study includes 1078 spousal dyads. In order to allow for longitudinal analysis and to rule out separation and re-marriage as a factor that shapes mental health, to be included in the sample, couples needed to be continuously married between 2006 and 2010. This allows us to control for the effect of marital transitions on mental health. Additionally, both spouses needed to complete the lifestyle and psychosocial questionnaire. Participants’ age was M = 65.12 (SD = 8.30) for husbands and M = 61.78 (SD = 8.65) for wives at T1. In total, 85% of participants described themselves as White, and had 13 years of education (M = 13.16, SD = 3.23 for men; M = 13.06, SD = 2.90 for women). Couples were married for 34.1 years on average (SD = 14.60), and for 67.8% of men, and 70.5% of women, they were in their first marriage. Mean annual income from all sources (including work income of each and total non-job incomes) was USD 89,224.44 (for more details, please see https://hrs.isr.umich.edu/about/how-to-use-this-site (accessed on 20 March 2024)).

### 2.2. Measures

#### 2.2.1. Anxiety Symptoms

Anxiety symptoms were assessed using five questions drawn from the Beck Anxiety Inventory (BAI) [43]. Responses ranged from 1 = never to 4 = most of the time. Responses that had more than two missing items were not included [44]. Internal reliability was good (husbands: M = 1.452, SD = 0.511, α = 0.80; wives: M = 1.488, SD = 0.538, α = 0.81), and items were averaged to create an overall anxiety score.

#### 2.2.2. Depressive Symptoms

Depressive symptoms were measured using a shortened version of the Center for Epidemiological Studies-Depression Scale (CES-D) [45] consisting of 8 symptoms. Participants were asked about the presence (1) or absence (0) of symptoms experienced frequently throughout the previous week. The scale aggregated the responses from these eight items, with a higher score indicating a greater number of depressive symptoms (range: 0–8).

#### 2.2.3. Cynical Hostility

The five-item Cook–Medley Hostility Inventory [46,47] was used to assess cynical hostility. Sample items include ‘I think most people would lie in order to get ahead’ and ‘No one cares much what happens to you’. Responses ranged from 1 = strongly disagree to 6 = strongly agree. All five items were averaged together. Responses were counted as missing if a respondent had more than three missing responses (husbands: M = 3.034, SD = 1.102, α = 0.80 and wives: M = 2.679, SD =1.068, α = 0.79).

#### 2.2.4. Covariates

Covariates included age, race (0 = Not White 1 = White), and self-rated health (1 = poor to 5 = excellent; husbands M = 3.44, SD = 1.02; wives M = 3.5; SD = 1.03 at T1; husbands = M = 3.31; SD = 1.00; wives = M = 3.41; SD = 1.00 at T2). We additionally tested the models for robustness by including education measured in years, length of marriage, whether it is respondents’ first marriages or not (measure in number of current relations), and total household income. See the description under the Participants section.

### 2.3. Analysis

The Actor-Partner Interdependence Model (APIM) and Actor-Partner Interdependence Moderation Model (APIMoM) were used to assess the hypotheses [48]. Actor-Partner Interdependence models are used to assess dyadic associations while accounting for the dependence that exists between the two partners (that between the “actor” [one partner] and “partner” [their partner]) [49]. To address H1, we regressed depression and anxiety on cynical hostility. To address H2, we added the interaction term between a husband’s and wife’s cynical hostility to the equation. Each dependent variable was tested in a separate model, and the analysis was performed three times: once on the 2006 dataset, once on the 2010 dataset, and then longitudinally, regressing the 2010 dependent variable on 2006 predictors, while controlling for the same dependent variable in 2006. To assess any significant moderation effects, we probed the spouse’s cynical hostility at one standard deviation above and below its mean. Race, self-rated health, and age were included as covariates in all models.

## 3. Results

Descriptive statistics and correlations are presented in Table 1. Husbands’ 2006 cynical hostility was positively correlated with their own anxiety and depressive symptoms in 2006 (r = 0.24, *p* < 0.001 for both) and in 2010 (r = 0.23, *p* < 0.001, r = 0.18, *p* < 0.001, respectively). Husbands’ 2006 cynical hostility was also positively correlated with their wives’ 2006 anxiety and depressive symptoms (r = 0.14, *p* < 0.001; r = 0.15, *p* < 0.001, respectively), and 2010 anxiety and depressive symptoms (r = 0.14, *p* < 0.001; r = 0.15, *p* < 0.001, respectively). Wives’ 2006 cynical hostility was also positively correlated with their own anxiety and depressive symptoms in 2006 (r = 0.25, *p* < 0.001; r = 0.21, *p* < 0.001, respectively) and in 2010 (r = 0.24, *p* < 0.001; r = 0.21, *p* < 0.001, respectively). Wives’ 2006 cynical hostility was also positively associated with their husbands’ anxiety and depressive symptoms in 2006 (r = 0.16, *p* < 0.001; r = 0.17, *p* < 0.001) and 2010 (r = 0.15, *p* < 0.001; r = 0.16, *p* < 0.001, respectively).

### 3.1. Cross-Sectional Results of APIM for T1 (2006)

Testing the model for anxiety, we found that the model had good fit to the data (CFI = 0.985, TLI = 0.958, RMSEA = 0.025). Two actor effects were found: husbands and wives who reported greater cynical hostility reported greater anxiety (husbands: *β* = 0.16, *p* < 0.001; wives: *β* = 0.20, *p* < 0.001; see Table 2). One partner effect was found: for husbands only, wives’ greater cynical hostility was associated with their husbands’ greater anxiety (*β* = 0.06, *p* = 0.05).

We followed up this initial APIM with an APIMoM (see Table 2). This model had a good fit (CFI = 0.985, TLI = 0.951, RMSEA = 0.025). However, the interaction term was not statistically significant, suggesting that the relationship between one’s own cynical hostility and anxiety is independent of their spouse’s cynical hostility.

The same procedure was followed for depressive symptoms (see Table 2). The model showed a good fit to the data (CFI = 0.995, TLI = 0.986, RMSEA = 0.016). The results also showed similar patterns: for both husbands and wives, actor effects were found, such that their own cynical hostility was significantly associated with depressive symptoms (β = 0.16, *p* < 0.001 for husbands; β = 0.12, *p* < 0.001 for wives). We also found one partner effect: husbands’ depressive symptoms were significantly associated with their spouse’s cynical hostility (β = 0.06, *p* < 0.05). No significant interactions were found (CFI = 0.0995, TLI = 0.986, RMSEA = 0.015).

### 3.2. Cross-Sectional Results for T2 (2010)

Testing the model for anxiety (Table 3), we found that, similarly to 2006, both husbands’ and wives’ own cynical hostility were significantly associated with anxiety, displaying actor effects s (*β* = 0.16, *p* < 0.001 for husbands; *β* = 0.20, *p* < 0.001 for wives). For husbands, spouse’s cynical hostility was significantly associated with anxiety (*β* = 0.09, *p* < 0.01), suggesting a partner effect. Husband’s cynical hostility was also marginally associated with their spouse’s cynical hostility (*β* = 0.05, *p* < 0.10). The model showed a good fit to the data (CFI = 0.974, TLI = 0.925, RMSEA = 0.037).

In the second step, we added the interaction term with APIMoM. A significant interaction was found between wives’ cynical hostility and husbands’ cynical hostility on wives’ anxiety (β = 0.10, *p* < 0.01). We probed this interaction one standard deviation below (low) and above (high) the mean level of husbands’ cynical hostility. The results suggest that when a husband’s cynical hostility is higher, the association between their wife’s cynical hostility and her own anxiety is stronger (b = 0.05, *p* < 0.05 when husband’s cynical hostility is low; b = 0.10, *p* < 0.001 when husband’s cynical hostility is at the mean; b = 0.15, *p* < 0.001 when husband’s cynical hostility is high).

The same model was tested for depressive symptoms (Table 3). For both husbands and wives, their own cynical hostility was significantly associated with their own depressive symptoms showing the expected actor effects (*β* = 0.10, *p* < 0.001 for husbands; β = 0.21, *p* < 0.001 for wives). We also found one partner’s effect, such that husbands’ depressive symptoms were also significantly associated with their spouse’s cynical hostility (β = 0.10, *p* < 0.001). The model showed a good fit to the data (CFI = 1.00, TLI = 1.00, RMSEA = 0.00).

For both husbands and wives, the interaction between one’s own and their spouse’s cynical hostility was significantly associated with depressive symptoms (β = 0.10, *p* < 0.001 for husbands; β = 0.09, *p* < 0.05 for wives). Probing the interaction, we found that for both husbands and wives, higher levels of spouse’s cynical hostility strengthen the association between one’s own cynical hostility and depressive symptoms (for husbands: b = 0.007, *p* = ns when spouse’s cynical hostility is low; b = 0.13, *p* < 0.01 when at the mean; b = 0.25, *p* < 0.001 when high. For wives: b = 0.18, *p* < 0.001 when husband’s cynical hostility is low; b = 0.31, *p* < 0.001 when at the mean; b = 0.44, *p* < 0.001 when high).

Furthermore, we examined whether the effects of one’s own and their partner’s cynical hostility may be longitudinally meaningful, showing enduring effects. To carry this out, we used T2 anxiety and depressive symptoms as dependent variables, regressing them on husbands’ and wives’ cynical hostility, while controlling for the same dependent variable at T1 and the T1 covariates. Results suggested that for both husbands and wives, only one’s own cynical hostility is a significant predictor of anxiety symptoms four years after first measurement (*β* = 0.09, *p* < 0.001 for husbands; *β* = 0.10, *p* < 0.01 for wives) (CFI = 0.986, TLI = 0.964, RMSEA = 0.035). We repeated the analysis for depressive symptoms. Only wives’ cynical hostility was predictive of their own depressive symptoms four years after the previous measurement (*β* = 0.11, *p* < 0.05) (CFI = 0.997, TLI = 0.993, RMSEA = 0.014).

Adding the interaction term to each of the equations, we found that for women, their husbands’ anxiety symptoms marginally moderate the association between own cynical hostility and anxiety symptoms (*β* = 0.06, *p* = 0.052), such that the effect of one’s own cynical hostility on anxiety is stronger when their husbands’ cynical hostility is higher (b = 0.02, *p* = ns when husband’s cynical hostility is low; b = 0.05, *p* < 0.01 when mean; b = 0.07, *p* < 0.001 when husband’s cynical hostility is high) (CFI = 0.985, TLI = 0.958, RMSEA = 0.036). None of the interactions were significant for the models predicting depressive symptoms (CFI = 0.997, TLI = 0.992, RMSEA = 0.014).

To compare the gender and actor/partner effects we ran the models again, each time constraining one path. We then used the Satorra–Bentler test to compare the χ^2^ fit indices. For each dependent variable, we ran five comparisons: (a) actor vs. partner’s effects for men; (b) actor vs. partner’s effect for women; (c) actor (own) effect for men vs. women; (d) partner’s effect for men vs. women; and (e) interaction effects for men vs. women. Results are graphically displayed (Figure 1a–d). (a) For men, actor and partner effects differed in 2006 only. (b) A consistent difference was found in all models between the effect of females’ own and females’ partners’ cynical hostility, suggesting that, for females, their own cynical hostility exerted a stronger effect than their partners’ cynical hostility. (c) No gender differences were found for the effect of own cynical hostility on neither anxiety nor depressive symptoms. (d) Similarly, in all but one comparison, no gender effects were found for a partner’s cynical hostility on neither anxiety nor depressive symptoms. One exception was inequality found for men’s and women’s partners’ effect on depressive symptoms in 2010. For this, the unconstrained model had a significant better fit, suggesting that the effect of a partner for men was stronger than for women (See Figure 1a–d).

Finally, we ran the models once more without the covariates, and once more with additional covariates, including education, length of the current relations, number of current relations, and income (divided by 100). Without the covariates, the pattern of results remained similar, with the exception of significant effects found for a partner’s effect on a wife’s anxiety and depression in 2006 (b = 0.07, se = 0.03, *p* < 0.05 and b = 0.19, se = 0.03, *p* < 0.001, respectively), and anxiety in 2010 (b = 0.07, se = 0.03, *p* < 0.05). With the additional covariates included, a husband’s partner effect of cynical hostility on anxiety and depressive symptoms became non-significant. Other effects remained stable, and none of the added covariates were consistently linked to the dependent variables.

## 4. Discussion

The current longitudinal dyadic study examined one’s own and a partner’s cynical hostility as potential predictors of anxiety and depressive symptoms. Previous studies suggest that one’s own cynical hostility has meaningful associations with physical and mental health [11,50], as well as with social relationships [2,4,25]. Some evidence also suggests that both partners’ cynical hostility may shape both partners’ health, wellbeing, and social relations [4,25,33]. In line with the previous findings, our results indicated that both husbands’ and wives’ mental health is negatively associated with cynical hostility, highlighting the adverse effects of cynical hostility on one’s anxiety and depressive symptoms; however, whereas for both partners, one’s own cynical hostility predicted anxiety and depressive symptoms—both within each time point, as well as longitudinally—partner’s cynical hostility predicted mental health only for husbands.

Men’s greater susceptibility to their spouse’s cynical hostility can be attributed to at least two mechanisms: First, it is possible that women are more expressive of their hostile attitudes; thus, they have a more prominent role in setting the relational atmosphere [51]. In this sense, it seems that overall, both partners’ mental health is impacted by the wife’s cynical hostility more than her male partner, suggesting that although the couple’s emotional experience is co-constructed [52], in the case of cynical hostility, it is the wife’s contribution that colors the couple interaction dynamics. Second, as women are usually kin keepers in families, men tend to be more dependent on their spouses for social interactions than the other way around. This means that, whereas women may have greater social independency and are, hence, less affected by their husband’s social schemas, men usually name their spouse as their confidant [53]; hence they may be more adversely affected by their wives’ schemas and behaviors [54].

Our results also highlighted the complex interdependence that exists between couple partners, pointing to moderating effects of partner’s cynical hostility on the association between one’s own cynical hostility and mental health, especially for women. Specifically, when husbands’ cynical hostility was high, the association between their wives’ cynical hostility and their wives’ own anxiety and depressive symptoms was stronger (albeit only in 2010). This finding aligns with the previous findings that indicate that female partners tend to be more attentive to their male partners’ characteristics than vice versa [40]; thus, experience enhances vulnerability—reflected in anxiety and depressive symptoms—when their partner expresses cynical hostility. Moreover, when both partners have high cynical hostility, they may find it difficult to establish and achieve emotional intimacy within the intimate relationship [37]. According to our findings, this shared contribution affects the female more than the male partner, particularly in terms of her anxiety, suggesting that male and female partners may perceive intimacy differently and experience different emotional and mental outcomes. These findings shed significant light on the intricate processes within couple dynamics, suggesting that an interpersonal examination is needed when exploring couple relationship characteristics.

### Limitations and Future Research

This study has several strengths but also some limitations that need to be acknowledged. One limitation of this study is its focus on the times of routine. Stressful times, such as COVID-19 or political tensions, may show similar or different trends of the explored associations; thus, studies should examine the role of cynical hostility in such contexts. Second, data in this study were based on self-reports. Collecting data concerning perceptions of one’s partner’s cynical hostility and employing observational data or daily diary design can assist in advancing the understanding of the possible effects of cynical hostility on one’s partners’ mental health. In addition, using different time frames to capture both short- and long-term implications would add to the research in this field. Finally, although the sample aspired to be as representative as possible, it may lack in this aspect; thus, future studies should look more closely at the associations between the variables studied in this paper within and across different cultural and social settings to gain further understanding of these associations. Along this line, the model should be studied among non-heterosexual couples to pinpoint any similarities or differences in terms of perceived gender roles, and also explore these on different age groups. 

Despite these limitations, this study suggests that cynical hostility is an important cognitive schema that does not only affect one’s own emotional reactions but also one’s spouse’s (and possibly other close others’) wellbeing. In this sense, one’s own and their spouse’s cynical hostility can, perhaps, have different effects in different situations. This understanding extends the theoretical frame on cynical hostility as an individual trait to the realm of interpersonal contexts that should be considered. Practically, the research reported here underscores the importance of examining both one’s own and their partner’s social cognitive schemas when consulting both individuals and couples. In this sense, interventions aimed at reducing cynical hostility could focus on reconstructing the cognitive schema of cynicism to change the way individuals and their partner perceive, interpret, and, in turn, react to others in interpersonal situations. Cognitive behavioral therapy programs (e.g., the Growing Pro-Social program, [55]; Imagery Enhanced Cognitive Restructuring, [56]) can be useful in reducing the negative attributes of individuals and couples.

## Figures and Tables

**Figure 1 behavsci-14-00283-f001:**
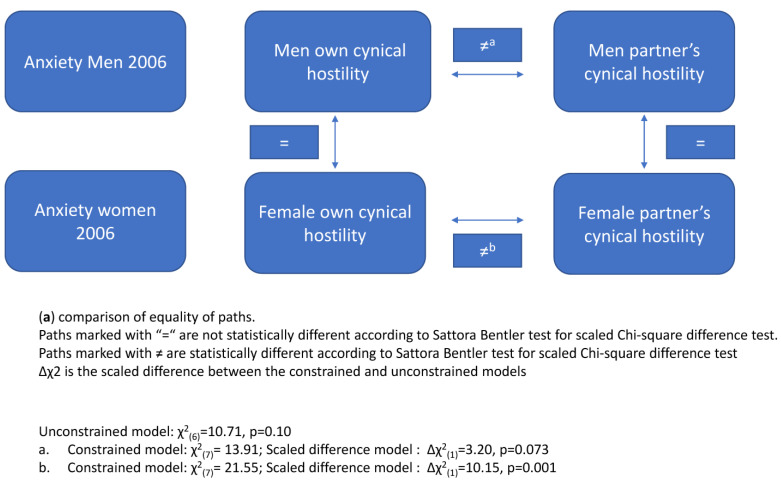
(**a**–**d**) comparison of equality of paths.

**Table 1 behavsci-14-00283-t001:** Descriptive statistics and correlations between study variables.

	M	SD	1	2	3	4	5	6	7	8	9	10	11
1. Husbands’ cynical hostility 2006	3.13	1.11											
2. Husbands’ anxiety 2006	1.46	0.51	0.24 ***										
3. Husbands’ depressive symptoms 2006	0.90	1.51	0.24 ***	0.39 ***									
4. Husbands’ cynical hostility 2010	3.02	1.11	0.59 ***	0.19 ***	0.22 ***								
5. Husbands’ anxiety 2010	1.46	0.52	0.23 ***	0.57 ***	0.35 ***	0.25 ***							
6. Husband’s depressive symptoms 2010	0.86	1.50	0.18 ***	0.34 ***	0.52 ***	0.18 ***	0.41 ***						
7. Wives’ cynical hostility 2006	2.72	1.08	0.33 ***	0.16 ***	0.17 ***	0.26 ***	0.15 ***	0.13 ***					
8. Wives’ anxiety 2006	1.52	0.53	0.14 ***	0.23 ***	0.16 ***	0.12 ***	00.17 ***	0.15 ***	0.25 ***				
9. Wives’ depressive symptoms 2006	1.21	1.81	0.15 ***	0.12 ***	0.23 ***	0.11 ***	0.08 **	0.18 ***	0.21 ***	0.44 ***			
10. Wives’ cynical hostility 2010	2.66	1.06	0.28 ***	0.14 ***	0.15 ***	0.28 ***	0.17 ***	0.17 ***	0.58 ***	0.25 ***	0.20 ***		
11. Wives’ anxiety 2010	1.50	0.55	0.14 ***	0.22 ***	0.11 ***	0.13 ***	0.27 ***	0.16 ***	0.24 ***	0.53 ***	0.36 ***	0.26 ***	
12. Wives’ depressive symptoms 2010	1.11	1.74	0.15 ***	0.15 ***	0.18 ***	0.09 ***	0.14 ***	0.21 ***	0.21 ***	0.36 ***	0.54 ***	0.26 ***	0.46 ***

** *p* < 0.01; *** *p* < 0.001.

**Table 2 behavsci-14-00283-t002:** Standardized coefficients (STDYXs) for the association between husbands’ and wives’ cynical hostility and mental health in 2006.

	Overall Model Anxiety ^1^		Moderation Effects Anxiety		Overall Model Depressive Symptoms ^2^		Moderation Effects Depressive Symptoms	
	Husbands	Wives	Husbands	Wives	Husbands	Wives	Husbands	Wives
	*β*/*SE*	*β*/*SE*	*β*/*SE*	*β*/*SE*	*β*/*SE*	*β*/*SE*	*β*/*SE*	*β*/*SE*
Wife Race		0.03/0.03		0.04/0.03		0.03/0.03		0.03/0.03
Husband Race	0.01/0.03		0.01/0.03		0.03/0.03		−0.03/0.03	
Wife Age		−0.04/0.03		−0.04/0.03		−0.13 ***/0.03		−0.13 ***/0.03
Husband Age	−0.04/0.03		−0.04/0.03		−0.11 ***/0.03		−0.11 ***/0.03	
Wife Self-Rated Health		−0.26 ***/0.03		−0.25 ***/0.03		−0.34 ***/0.03		−0.34 ***/0.03
Husband Self-Rated Health	−0.25 ***/0.03		−0.25 ***/0.03		−0.30 ***/0.03		−0.30 ***/0.03	
Wife Cynical Hostility (WCH)	0.06 */0.03	0.20 ***/0.03	0.07/0.08	0.15+/0.08	0.06 */0.03	0.12 ***/0.03	0.03/0.06	−0.07/0.07
Husband Cynical Hostility (HCH)	0.16 ***/0.03	0.04/0.030	0.06 */0.08	−0.003/0.07	0.16 ***/0.03	0.05/0.03	0.12+/0.07	−0.01/0.08
WCH × CH			−0.01/0.13	0.08			0.06/0.12	0.20/0.13
Model Fit	RMSEA = 0.025, CFI = 0.985, TLI = 0.985		RMSEA = 0.025, CFI = 0.985, TLI = 0.981		RMSEA = 0.016, CFI = 0.995, TLI = 0.986			

Note: * *p* < 0.05; *** *p* < 0.01. ^1^ Model fit: *CFI* = 0.952, *SRMR* = 0.025. ^2^ Model fit: *CFI* = 0.957, *SRMR =* 0.024.

**Table 3 behavsci-14-00283-t003:** Standardized coefficients (STDYXs) for the association between husbands’ and wives’ cynical hostility and mental health in 2010.

	Overall Model Anxiety		Moderation Effects Anxiety		Overall Model Depressive Symptoms		Moderation Effects Depressive Symptoms	
	Husbands	Wives	Husbands	Wives	Husbands	Wives	Husbands	Wives
	*β*/*SE*	*β*/*SE*	*β*/*SE*	*β*/*SE*	*β*/*SE*	*β*/*SE*	*β*/*SE*	*β*/*SE*
Wife Race		0.04/0.03		0.04/0.03		0.006/0.03		0.01/0.03
Husband Race	0.01/0.03		0.01/0.03		−0.03/0.03		−0.03/0.03	
Wife Age		0.03/0.03		0.03/0.03		−0.03/0.03		−0.03/0.03
Husband Age	−0.02/0.03		−0.02/0.03		−0.06 */0.03		−0.05+/0.03	
Wife Self-Rated Health		−0.25 ***/0.03		−0.25 ***/0.03		−0.29 ***/0.03		−0.28 ***/0.03
Husband Self-Rated Health	−0.31 ***/0.03		−0.31 ***/0.03		−0.25 ***/0.03		−0.24 ***/0.03	
Wife Cynical Hostility (WCH)	0.09 **/0.03	0.20/0.03 ***	0.09 **/0.03	0.19 ***/0.03	0.09 **/0.03	0.21 ***/0.03	0.09 **/0.03	0.20 ***/0.03
Husband Cynical Hostility (HCH)	0.16 ***/0.03	0.05+/0.03	0.16 ***/0.03	0.04/0.03	0.10 **/0.03	−0.01/0.03	0.10 **/0.03	−0.01/0.03
WCH × HCH			0.03/0.03	0.10 **/0.03			0.10 **/0.04	0.09 */0.03
Model Fit	RMSEA = 0.037, CFI = 0.974, TLI = 0.925		RMSEA = 0.035, CFI = 0.977, TLI = 0.927		RMSEA = 0.000, CFI = 1.00, TLI = 1.00		RMSEA = 0.000, CFI = 1.00, TLI = 1.00	

Note: * *p* < 0.05; ** *p* < 0.01, *** *p* < 0.001.

## Data Availability

The data used in this research are publicly available (pre-registration is required) (see: https://hrs.isr.umich.edu/about, accessed on 20 March 2024).

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
