# Peer review of "Links between Couples’ Cynical Hostility and Mental Health: A Dyadic Investigation of Older Couples"

_behavsci, 2024, doi:10.3390/bs14040283_

Round 1
Reviewer 1 Report
Comments and Suggestions for Authors...

Author Response
Thank you very much for your positive feedback on our manuscript!
Reviewer 2 Report
Comments and Suggestions for Authors
The abstract is very well written because it gives a global overview of the proposal. In the keywords you should change ‘couples’ to ‘older couples’.
The theoretical framework is consistent and supported by several authors, which allows the appreciation of different points of view. There could be more recent quotes, it turns out that some of the quotes are a few years old. However, very current quotes also appear which balance this aspect. The empirical part is well justified, and the statistical methods used agree with the hypotheses formulated. Presenting data in tables also makes data analysis easier to understand.
In the discussion of the results, different authors also were mobilized, which supports the reflection and conclusions, which in my perspective should be praised.
Another aspect that I highlight positively has to do with the limitations of the study, which safeguards an unlimited generalization of the conclusions. And, at the same time, they present proposals for future investigations, which confirms an in-depth critical reflection.
Author Response

(The authors gave the same response as above.)

Reviewer 3 Report
Comments and Suggestions for Authors
This study of American a sample of persons aged 65 and older from 2006 and 2010 waves of the Health and Retirement Study examined the relationship of clinical hostility and anxiety and/or depressives symptoms.
The introduction provides some evidence to support the study but lacks support for the limited number of covariates included, and omits important variables such as length of time married at each time period, number of prior marriages (before 2006), support networks (formal and informal), and gender identity. Also lacking are educational level and income, known social determinants of health that effect both depression and anxiety symptoms in older adults.
Sullivan’s interpersonal theory of personality may a good model to help understand the major variables of this study, but it lacks application to explain the cohort variability of older adults’ personalities and cumulative interpersonal experiences of a lifetime. The authors also seem to assume that personality characteristics such as cynical hostility are permanent and unchanging, not responsive to more than just interpersonal relationships of a marriage. This lack of understanding of the major psychological and social theories of aging undermines the impact of this study.
Although the design and method of the study are clear and presented well, the lack of a table of demographic covariates makes interpretation of the findings difficult and questionable.
This sample is quite young and racially does not represent the population of older adults in the United States as suggested. This is a significant limitation that needs to be included in the discussion section. Also please explain the term “Actor-Partner” first introduced in Analysis section. If you mean the partners self-identified as wife or husband, you cannot assume this also means gender, only role of wife or husband, and your results and discussion should reflect this role versus gender. Roles are not personality characteristics but learned through culture as well as interactions, and this also changes with older adult cohorts.
The conclusions call for more research, but I ask, to what end? Do your results support Sullivan’s model? If yes or no, give more details. Can you potentially change any of the variables in this study with better psychological interventions? If yes, offer some ideas for future study. Unfortunately, as currently written, this study does not add to our understanding of older adults, perhaps of the Sullivan model, but again, I do not see how this helps our understanding of older adults.
Author Response
Reviewer 3:
- This study of American a sample of persons aged 65 and older from 2006 and 2010 waves of the Health and Retirement Study examined the relationship of clinical hostility and anxiety and/or depressives symptoms. The introduction provides some evidence to support the study but lacks support for the limited number of covariates included, and omits important variables such as length of time married at each time period, number of prior marriages (before 2006), support networks (formal and informal), and gender identity. Also lacking are educational level and income, known social determinants of health that effect both depression and anxiety symptoms in older adults.
Response: Following the above comment, we ran the analyses again, while including length of current relations, the number of the current marriage, education and income. Other than income, that was linked to anxiety in 2010, none of the other covariates showed any consistent significant correlations with the dependent variables. Moreover, with the exception of two weak and marginally significant effects that turned non-significant, the pattern of results remained similar. Thus, we did not alter the text on our findings but rather, we now added some text to indicate we included these variables in our analyses to test for robustness (lines 179-182; 303-305; 308-311).
- Sullivan’s interpersonal theory of personality may a good model to help understand the major variables of this study, but it lacks application to explain the cohort variability of older adults’ personalities and cumulative interpersonal experiences of a lifetime. The authors also seem to assume that personality characteristics such as cynical hostility are permanent and unchanging, not responsive to more than just interpersonal relationships of a marriage. This lack of understanding of the major psychological and social theories of aging undermines the impact of this study.
Response: We thank the reviewer for pointing this out. We now added text to support the assertion that cynical hostility is more of a personality trait, that starts forming in early life and remains stable over time, than influenced by later psychological influences or life events (see lines 38-42, 108, 110). However, following the reviewer's comment, we now also acknowledge the possible role of social environmental factors, such as income and educational level, in the potential variability of cynical hostility among older adults and include such covariates in our analyses (see lines 40-42, 116-121, 132-134).
- Although the design and method of the study are clear and presented well, the lack of a table of demographic covariates makes interpretation of the findings difficult and questionable.
Response:
We added demographic information to the Participants subsection (lines 147-153). We also suggest the readers to look into the Health and Retirement Study website for more detailed information about the sampling method and demographic information.
- This sample is quite young and racially does not represent the population of older adults in the United States as suggested. This is a significant limitation that needs to be included in the discussion section.
Response: The sample included older adult couples with a mean age of over 61 (over 65 among male participants). Among men, 43% were over 65, and among women 40% were over 65. Although not very old, the sample still consists of older adult couples and provides significant information about the associations between the study variables among older population. In addition, the HRS project, which we used its sample for this study, aspires to reach a good representation of older adults in the US and does a pretty good job at this aspect. We do, however, suggest in the Limitation section for future studies to look closer at different cultural groups in the US and elsewhere to gain further understanding of the studied phenomenon across different cultural settings (see lines 362-371). We also refer the readers to the very detailed sampling reports available at the HRS website.
- Also please explain the term “Actor-Partner” first introduced in Analysis section. If you mean the partners self-identified as wife or husband, you cannot assume this also means gender, only role of wife or husband, and your results and discussion should reflect this role versus gender. Roles are not personality characteristics but learned through culture as well as interactions, and this also changes with older adult cohorts.
Response: We now explain "actor-partner" in the analysis section (lines 185-188). The distinction in our study is based on gender. Gender identity or identification with the role of husband or wife were not measured in the HRS. However, based on previous studies on heterosexual couples using the HRS and our knowledge as gerontologists, we believe that gender and gender-role in this sample should be extremely close. We acknowledge, however, that future studies could use samples of non-heterosexual couples and couple from diverse backgrounds (lines 362-371, specifically, 366-367).
- The conclusions call for more research, but I ask, to what end? Do your results support Sullivan’s model? If yes or no, give more details. Can you potentially change any of the variables in this study with better psychological interventions? If yes, offer some ideas for future study. Unfortunately, as currently written, this study does not add to our understanding of older adults, perhaps of the Sullivan model, but again, I do not see how this helps our understanding of older adults.
Response: Practically, we now detail that certain interventions addressing cognitive schemas that were found useful, and suggest to consider these in couples' therapy (lines 379-384). Theoretically, we suggest that understanding older adults' mental health may require taking into account not only their own trait and cognitions, but also their spouse's (e.g., lines 375-377).
Round 2
Reviewer 3 Report
Comments and Suggestions for Authors
I applaud the authors for their excellent comments back to me. All concerns have been addressed.